# Catalysis with Silver: From Complexes and Nanoparticles to MORALs and Single-Atom Catalysts

**Mario Pagliaro [1], Cristina Della Pina [2,*], Francesco Mauriello [3] and Rosaria Ciriminna [1,*]**

1   Istituto per lo Studio dei Materiali Nanostrutturati, CNR, via U. La Malfa 153, I-90146 Palermo, Italy; mario.pagliaro@cnr.it
2   Dipartimento di Chimica, Università degli Studi di Milano, via Golgi 19, I-20133 Milano, Italy
3   Dipartimento DICEAM, Università Mediterranea di Reggio Calabria, Loc. Feo di Vito, I-89122 Reggio Calabria, Italy; francesco.mauriello@unirc.it
*   Correspondence: cristina.dellapina@unimi.it (C.D.P.); rosaria.ciriminna@cnr.it (R.C.)

 

**Abstract:** Silver catalysis has a rich and versatile chemistry now expanding from processes mediated by silver complexes and silver nanoparticles to transformations catalyzed by silver metal organic alloys and single-atom catalysts. Focusing on selected recent advances, we identify the key advantages offered by these highly selective heterogeneous catalysts. We conclude by offering seven research and educational guidelines aimed at further progressing the field of new generation silver-based catalytic materials.

**Keywords:** silver; metal-organic alloys (MORAL); silver-based catalytic materials; single-atom catalysis

## 1. Introduction

From cycloaddition and cyclization through fluorination, from C–C cross coupling reactions to reductions and oxidations, catalytic processes mediated by silver(I) complexes and silver nanoparticles (NPs) are numerous and increasingly employed in synthetic organic chemistry [1–12].

Silver(I) is a π-Lewis acid suitable for activation of carbon-carbon multiple bonds [3–5]. To absorb visible light, plasmonic resonant silver nanoparticles are used for the photocatalytic activation of $O_2$ in visible-light oxidations and are a highly promising new means for environmental decontamination, including removal of water disinfection by-products [6].

So wide and rich is the catalysis chemistry of silver, that a recent comprehensive book on the topic had to be split into two volumes exceeding 1000 pages [1]. Silver catalysts, furthermore, are employed in the chemical industry in a few processes of great commercial relevance.

Produced in amounts close to 30 million tons annually, ethylene oxide (EO) is an essential building block of the chemical industry manufactured via epoxidation of ethylene with oxygen mediated by a 15 wt % silver catalyst comprised of large silver NPs (100–200 nm) supported on low-surface-area alumina (with alkaline species as promoters to enhance the selectivity to EO) [7].

Silver in gauze shape is used industrially as a catalyst in the oxidative dehydrogenation of methanol to formaldehyde at 600 or 700 °C ("methanol ballast process" in which only air and methanol are fed into the reactor without extra water in the reactant mixture, and the BASF process, respectively, where BASF is the chemical company developing this process). The BASF process results in 97–98% methanol conversion with an aldehyde yield between 89.5 and 90.5% [8]. Alone, the EO synthesis in 2015 saw a 103% increase in the amount of silver employed in the world's EO plants compared with 2014, with about 3900 tons of silver residing in said plants [9].

In the last two decades, significant progress has occurred in virtually all areas of catalysis science and technology. The reader is referred to the aforementioned recent book [1], as well as to excellent

reviews on heterogeneous silver nanoparticle catalysis for the synthesis of natural products and pharmaceutical ingredients [10], and on Ag(I) complexes in the catalytic modification of biomolecules such as oligosaccharides and peptides [11]. Since then, progress has continued at a fast pace also in the latter two areas. For instance, scholars in Spain reported in 2017 an important advance in the selective oxidation of ethylene to EO based on different silver nanostructures dispersed on a tubular copper oxide matrix leading to high yields at lower reaction temperatures with a single pretreatment step at ambient pressure in contrast with the common practice of continuously feeding organochlorinated promoter precursors during the reaction [12]. Currently, for instance, a themed issue on silver catalysis focusing on synthetic organic chemistry including asymmetric catalysis and the synthesis of active pharmaceutically ingredients is being published in the catalysis journal of the European chemical societies [1]. In this study, we focus on selected recent advances concerning processes mediated by new metal organic alloys (MORALs) and silver single-atom catalysts (SACs). First reported in 2002 with the entrapment of organic dyes in metallic silver [13], MORALs are hybrid materials with immense applicative potential in all areas where metals are used, including catalysis [14,15]. Single-atom catalysts, in their turn, are a potentially disruptive chemical technology [16], now approaching practical application for the production of valued bulk and fine chemicals [17].

Catalysis scholars are well acquainted with the importance of metal particle size in the catalytic performance. Minimizing the size to the nanoscale and below has represented a breakthrough in catalysis over the last two decades. Numerous studies demonstrated that downsizing the nanoparticle size to sub-nanoscale can either improve the catalytic activity and selectivity in various oxidation and reduction reactions, or even render highly reactive renowned inert metals such as gold [18–20]. Nevertheless, sub-nanoclusters still present multiple active centers some of which can promote undesirable side reactions. Developing stable solid catalysts with well-defined, atomically dispersed active centers, namely SACs, is perhaps the most important achievement in catalysis research of the last decade [16]. Scheme 1 displays the effects of minimizing Ag particle size on the catalytic activity in terms and on surface free energy.

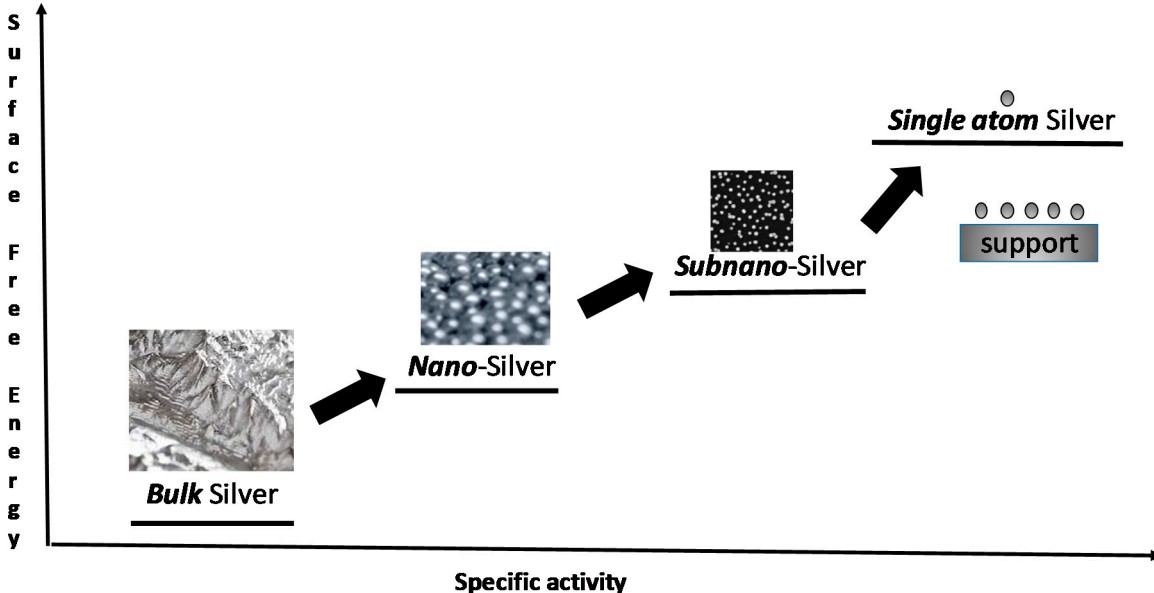

**Scheme 1.** Effect of minimizing Ag particle size on catalytic specific activity. [Images of nano- and subnano-silver modified from Ref. [21]].

In this study, we identify the new possibilities and the key advantages offered by new MORAL and single-atom silver catalysts through selected examples. In other words, rather than providing a comprehensive review of all achievements in catalysis with these new materials, we offer a critical insight aimed at suggesting avenues to guide the development of new single-atom and silver-organics

alloyed catalysts. Several processes of high relevance in tomorrow's chemical industry, we argue in the following, will be catalyzed by these new generation silver catalysts.

## 2. Enhanced Catalysis with Silver MORALs

Catalysis was the first field of application of molecularly doped metals introduced in 2002. Three years later, Avnir and co-workers reported that the entrapment of the polyacid Nafion within silver affords a catalyst that showed excellent performance in the acid-catalyzed pinacol–pinacolone rearrangement and in the dehydration of 2-phenylethanol to styrene [22] (Scheme 2).

Rearrangement of *pinacol* to *pinacolone* by ***Nafion@Ag catalyst***

Dehydration of *phenylethanol* to *styrene* by ***Nafion@Ag catalyst***

**Scheme 2.** Catalytic conversion of pinacol to pinacolone, and catalytic dehydration of phenylethanol to styrene. [Reprinted from Ref. [22], copyright (2005) with kind permission from Wiley].

Underlining how "the metal's sea of electrons is a friendly environment for the various positively charged species formed along the reaction mechanism" [22] starting with diffusion of the reactants through the inner porosity of the superacid Nafion@Ag followed by product diffusion out of the porous material into the solution, the team achieved high product yields of 80% and 90% for both rearrangement and dehydration reactions, respectively.

In 2007, Avnir, Grader, and co-workers reported a surprising new result: silver doped with the organic dye Congo-red (CR@Ag) is a far better catalyst for methanol oxidation to formaldehyde when compared to pure silver [23]. The hybrid catalyst lowers the temperature, needed to reach maximal conversion to formaldehyde by more than 100 °C, and by 200 °C when it was necessary to reach the maximal selectivity (aldehyde formation), and increasing the maximal space velocity by a factor of two. The example is particularly relevant here because advances achieved with this new catalyst nicely render the potential of heterogeneous catalysis with silver MORALs.

The catalyst is prepared according to a simple, green and highly reproducible reduction process carried out in water only (Equation (1)), by mixing a 125-mL water solution of 0.018 mol $AgNO_3$, 0.017 mol sodium hypophosphite, and 0.14 mmol of CR, which was mechanically stirred at room temperature for 4 days [23]:

$$2AgNO_3(aq) + NaH_2PO_2(aq) + CR(aq) + H_2O(l) \rightarrow CR@(2Ag)(s) + NaH_2PO_3(aq) + 2HNO_3(aq) \quad (1)$$

The thermogravimetric and differential thermal analysis coupled to mass spectroscopy of CR@Ag under an oxidative atmosphere, showed that Ag strongly catalyzes the CR oxidation, with full decomposition of CR (and loss of $CO_2$, NO and $SO_2$) occurring at ca. 500 °C:

Above this temperature, only carbon and sodium carbonate coexist with the Ag, and evidently this composite was stable in the methanol–air mixture, allowing steady-state conditions to be reached easily.

The catalytic reaction was carried out at different temperatures, gradually decreasing from 500 to 200 °C, by employing the "methanol ballast process" version using the CR@Ag catalyst after thermal treatment (heating of the fresh catalyst in air at 200 °C for 2 h followed by its heating at 5 °C min$^{-1}$ rate to 500 °C, keeping this temperature for 0.5 h).

The team made the hypothesis that the carbonized organic dopant activates the surface of the Ag by effective reduction of the Ag oxide interface, with the graphitized material acting as an adsorbing layer, concentrating methanol at the Ag surface.

Four years later, the team reported that the organic dopant increased the catalyst surface area from 600–3000 cm$^2$ g$^{-1}$ for undoped silver to 45,800 cm$^2$ g$^{-1}$ for CR@Ag prepared after 8 days of aging, whereas oxygen chemisorption goes from 32 cm$^2$ g$^{-1}$ for Ag to 893 cm$^2$ g$^{-1}$ for CR@Ag [24]. They also found that the optimal catalyst synthesis at room temperature required 8 (and not 4) days to allow the CR molecules to more deeply penetrate into the silver crystal aggregates. The large increase in the MORAL powder grain size after the methanol oxidation process (Figure 1) was due to the solid-state sintering during the catalytic procedure at 500 °C [23,24].

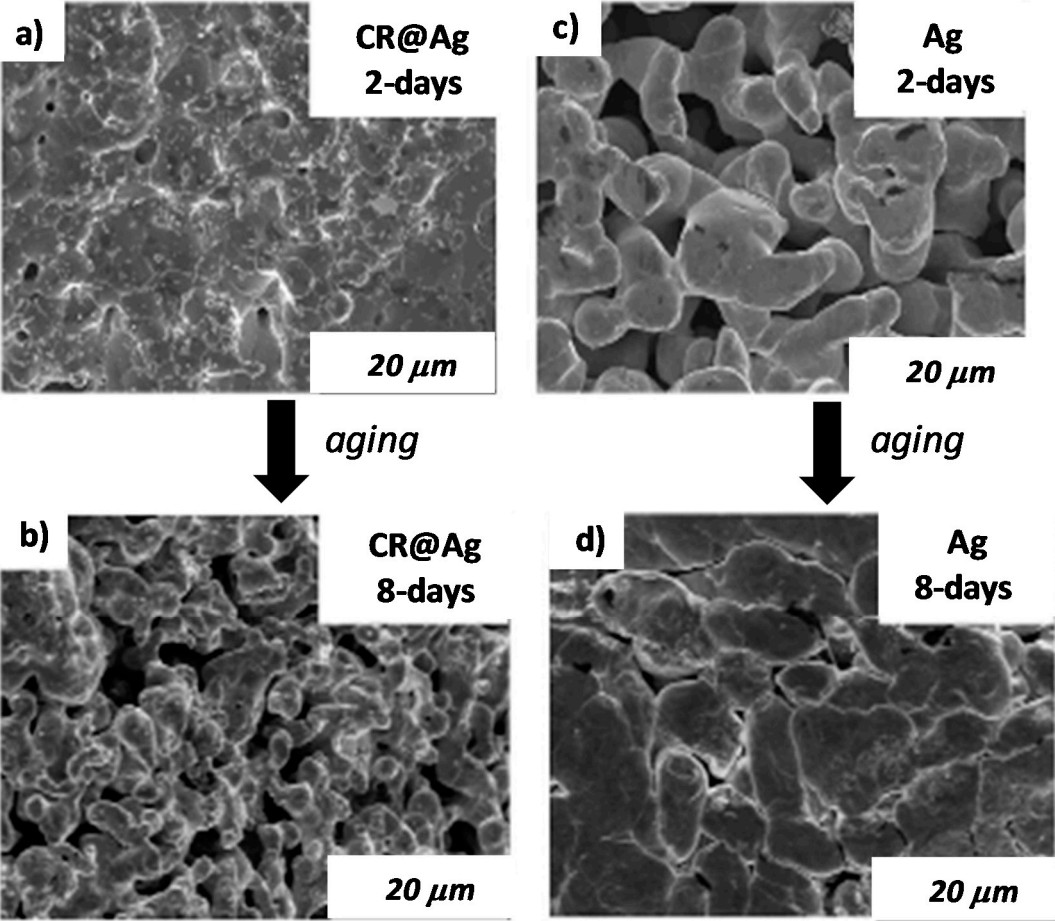

**Figure 1.** The morphology of CR@Ag (**a**,**b**) and of Ag (**c**,**d**) after the catalytic procedure: (**a**,**c**) 2-days catalyst synthesis time; (**b**,**d**) 8-days. [Reprinted from Ref. [24], copyright (2011) with kind permission from Royal Society of Chemistry, permission conveyed through Copyright Clearance Center, Inc.].

When the CR molecules were mainly placed on the outer surface of silver crystallites, as it happens in the 2-day and 4-day catalyst synthesis times, they did not prevent the sintering of the particles and served as a barrier to chemisorption of reagent molecules on the active centers of the silver agglomerates. On the other hand, when the CR residues were mainly located within the silver agglomerates as it happens in the 8 days aging synthesis, they served as an efficient barrier for the silver sintering process, leading to smaller catalyst particles with an increased number of active catalytic centers at the silver grain boundaries, a more porous morphology, and higher specific surface area, eventually leading to enhanced catalytic activity of doped silver [23], even though the high reaction temperature resulted in overall sintering of the catalyst particles, leading to a catalyst lifetime of one cycle only.

In 2009, Pagliaro and co-workers reported the first example of nanosized metal-organic alloys (nanoMORALs) by describing the synthesis of Ag nanoparticles doped with Cu(II) and Fe(III) phthalocyanines synthesized by reducing Ag(I) ions with $NaH_2PO_2$ in the presence of sodium dodecyl sulphate micelles [25]. Silver nanoparticles, indeed, are highly hydrophobic and this allows, for instance, the synthesis of elegant hollow and bimetallic nanostructures [26]. Beyond being significantly stabilized by the entrapment in the Ag nanoparticles, the photoactive phthalocyanines did not leach in water or in common organic solvents upon prolonged (4 days) immersion in the liquid phase. The presence of entrapped phthalocyanines did not influence the position and shape of typical lines in the XRD diffractogram ($2\theta$ = 38.28, 44.48, 64.68, 77.68, 81.88, 97.98) with similar patterns obtained both for the doped and undoped Ag nanoparticles [25]. The Scherrer's equation applied to all the six peaks leads to an average value of 11 nm of primary particle size, both for doped and undoped silver. Furthermore, SEM images highlight a similar morphology for both doped MORALs and undoped silver with aggregates of hundreds of nanometers up to few micrometers [25].

A few years later, the same Israeli team led by Avnir reported that nanoMORALs, 10–30 nm in size, of thermally activated CR@Ag deposited on $TiO_2$ nanofibers outperformed the catalytic performance of unsupported CR@Ag with respect to all the parameters of the oxidation of methanol to formaldehyde, including the weight hourly space velocity (18 times higher compared to unsupported CR@Ag), the space time yield (10 times higher), the reaction conversion, and the reaction selectivity with the fully activated catalyst in the second run, affording a conversion of 98% and selectivity of 85% [27]. The high efficiency of this novel type of triple hybrid-organic, metallic, and ceramic-catalyst (CR@Ag/$TiO_2$-nf) was due to the uniform spreading of the CR@Ag nanoparticles over the porous titania nanofibers, which results in higher porosity, larger surface area, and lower sinterability of the separated doped silver particles [27].

The catalyst after 2 h of preheating at 300 °C showed (Figure 2) a moderate increase in the particle size with the partially decomposed CR molecules remaining entrapped in the Ag nanocrystals, whereas the second gradual oxidative step caused a progressive degradation of CR resulting in diffusion and coalescence of the small silver crystallites into larger crystals. This first run produced the needed high surface area both by the formation of the carbonaceous residues and by the opening of the porosity of the Ag nanoparticles through the emission of small gaseous molecules leading both to higher local methanol concentration and to easier diffusion of $O_2$ into the silver nanocrystals. After the second run, the catalyst retains and even increases the high conversion and selectivity values achieved after the second crucial run, providing a catalyst with large applicative potential to replace the conventional Ag gauze employed on the methanol ballast process. The TEM images coupled with EDAX show the nanoMORAL particles ranging from ca. 30 nm down to a fraction of nanometers.

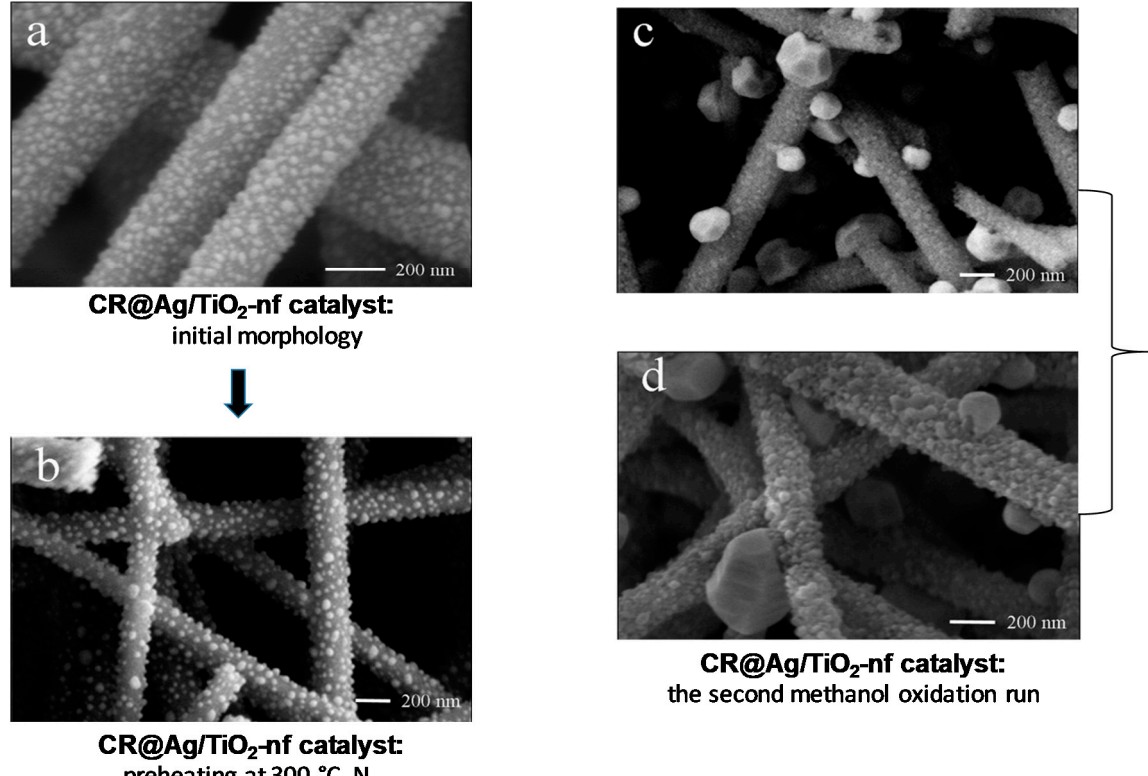

**Figure 2.** Morphology of 1.00%CR@Ag/TiO$_2$-nf catalyst: (**a**) initial morphology; (**b**) after preheating at 300 °C under N$_2$; (**c**,**d**) after the second methanol oxidation run. [Reprinted from Ref. [27], copyright (2013) with kind permission from American Chemical Society].

## 3. Single-Atom Silver Catalysts

In 2012, Tang, Li, and co-workers in China reported the discovery that a single-atom Ag catalyst (SAC) easily obtained by a simple thermal treatment of Ag NPs supported on Hollandite-type manganese oxide (HMO) nanorods shows an almost 10-fold higher catalytic activity in the oxidation of formaldehyde at low temperatures ($T < 80$ °C) to CO$_2$ and water when compared to the nanoparticle catalyst: 125 h$^{-1}$ turnover frequency (TOF) vs. 18 h$^{-1}$, respectively [28]. In detail, the Ag-HMO single-atom catalyst was obtained by annealing the Ag/HMO at 500 °C in air for 6 h. The Ag/HMO precursor was obtained by drying at 80 °C for 24 h a solid easily obtained by mixing an [Ag(NH$_3$)$_2$]OH solution and a H$_2$O$_2$ solution (30 wt.%, 30 mL) with a suspension (100 mL) containing the HMO (2.0 g) under stirring at 20 °C for 1 h [28].

Showing the versatility of single-atom silver catalysis, another Chinese team lately showed that the same stable single-atom silver catalyst (Ag-HMO) was able to mediate the photocatalytic reduction of CO$_2$ to CH$_4$ at ambient temperature and pressure with H$_2$ only as a reducing agent under visible light [29]. The catalyst outperformed the catalytic performances of Ag/P25, of Ag NPs deposited on the same hollandite-type manganese oxide nanorods (Ag/HMO), as well as of several other photocatalysts including Pt/TiO$_2$ under UV radiation (Table 1) [29]. For comparison, the activity of single-atom catalyst with the 30% silver loading reached a 6.0 μmol h$^{-1}$ g$^{-1}$ catalyst production rate for methane, a value > 33% higher than that of Ag/HMO.

**Table 1.** Production rate for $CH_4$ via $CO_2$ reduction with $H_2$ over different photocatalysts under different light radiation. [Modified from Ref. [29]].

| Catalyst | $CH_4$ Production Rate ($\mu mol\ h^{-1}\ mg^{-1}$ Catalyst) | Light Source |
|---|---|---|
| Ag-HMO | 6.0 | Visible light |
| $Rh/Al_2O_3$ | 0.60 | LED |
| $In_2O_3$-X(OH)y | 0.60 | Visible light |
| $TiO_2$ (P25) | 4.00 | UV |
| BiOCl/Zn-Cr | 1.60 | Sunlight |
| $TiO_2$ | 1.50 | Sunlight |
| $g$-$C_3N_4$/$SnS_2$ | 0.60 | Visible light |
| $Pt/TiO_2$ | 5.80 | UV |
| $Ru/TiO_2$ | 0.25 | UV |

The enhancement of photocatalytic $CO_2$ reduction activity was ascribed to the efficient electron transfer from photoexcited Ag atoms and NPs to the conduction band of the HMO semiconductor, and to the activation of adsorbed $H_2$ via transfer of the remaining holes in the Ag atoms conversed ($2H_2 + 4h^+ \rightarrow 4H^+$) providing the protons required along with the photoelectrons for the reduction of $CO_2$ (Equation (2)) [29]:

$$CO_2 + 8H^+ + 6e^- \rightarrow CH_4 + 2H_2O \tag{2}$$

It is also relevant from the practical application viewpoint, that the recent discovery of Zeng and co-workers that Ag single atoms ($Ag_1$) easily form on nanosized $\gamma$-$Al_2O_3$, in contrast to Ag nanoparticles on microsized $\gamma$-$Al_2O_3$ [30]. Knowing that $Ag^+$ has high electron affinity [31], the team made the hypothesis that Ag species are mainly anchored by the Al-OH sites of $\gamma$-$Al_2O_3$ through interaction between $Ag^+$ and O atoms. To verify the hypothesis, they employed surface-science measurements including magic angle spinning nuclear magnetic resonance (MAS NMR), in situ diffuse reflectance infrared Fourier transform (DRIFT) spectroscopy, high-angle annular dark-field scanning transmission electron microscopy (HAADF-STEM), and quantum chemical calculations.

Experiments and theory both suggest that terminal hydroxyl groups, abundant on the surface of nano-$\gamma$-$Al_2O_3$ comprised of nanoparticles of 10 nm average size, anchor and stabilize the $Ag_1$ species through interaction between Ag and the O atom in Al-OH terminal groups [30]. As a result, the $Ag_1/Al_2O_3$ catalyst with 1% Ag loading showed a far better catalytic performance than Ag/micro-$Al_2O_3$ for the selective catalytic reduction (SCR) of NO for both $C_3H_6$-$H_2$-SCR (NO 800 ppm, $C_3H_6$ 1565 ppm, $H_2$ 1%, $O_2$ 10%) and ethanol-SCR (NO 800 ppm, $C_2H_5OH$ 1565 ppm, $O_2$ 10%, $H_2O$ 5%) reactions between 200 and 500 °C.

Most importantly, investigation by HAADF-STEM of the $Ag_1$/nano-$Al_2O_3$ catalyst after HC-SCR of NO testing showed that the single-atom Ag species on the nano-alumina did not agglomerate, pointing to a high stability of Ag single atoms dispersed on the nano-alumina. In agreement with the DRIFT measurements, the quantum calculations confirmed that the (100) surfaces of $\gamma$-$Al_2O_3$, that are predominant in nanosized $\gamma$-$Al_2O_3$ and present only terminal hydroxyls on the surface, spontaneously afford formation of the single-atom $Ag_1/Al_2O_3$ catalyst via binding of the single Ag atoms to said terminal hydroxyls, chiefly via a highly stable (adsorption energies of Ag of 5.69 eV) staple-like structure formed by consuming two or three terminal hydroxyls. It is also remarkable, in sight of practical applications, the simplicity of the single-atom catalyst preparation that simply consists of stirring for 2 h the appropriate amount of nanosized $\gamma$-$Al_2O_3$ with aqueous $AgNO_3$, followed by water removal via vacuum rotary evaporation, drying at 105 °C overnight, and calcination at 500 °C for 3 h in air.

The practical relevance of these findings is better understood considering that the ethylene epoxidation catalyst is indeed comprised of $Ag/Al_2O_3$ and that Ag nanoparticle sintering was identified in 2004 as the main source for deactivation of the commercial catalyst [32], leading scholars to suggest

the development of a new generation of Ag-based catalysts working at lower reaction temperatures, since raising the temperature in order to keep the production of EO at a constant level results in faster catalyst deactivation and, even more importantly, in lower selectivity to EO. It is likely the new single-atom catalyst deposited on nanosized $\gamma$-Al$_2$O$_3$ might outperform the commercial nanoparticle catalyst, not only enabling lower reaction temperatures than the 260–280 °C. typically employed in ethylene epoxidation [32], but fully retaining its selective activity due to retention of the single-atom catalyst nature demonstrated for the Ag$_1$/Al$_2$O$_3$ catalyst up to 500 °C [30].

## 4. Research and Educational Consequences

A selected review of recent progress with new generation silver catalytic materials comprised of single-atom and silver-organics catalysts has several research and educational lessons to teach.

First, switching from conventional silver surfaces to silver-organics and silver single atoms greatly enhances the activity and the selectivity of silver-based catalysts not only in synthetic processes of interest in organic chemistry and in the fine chemicals industry, but also in processes of significant commercial relevance employed in the petrochemical industry such as methanol oxidation to formaldehyde and ethylene epoxidation to ethylene epoxide.

Second, silver-organic alloys and single-atom silver catalysts expand the versatility and applicability of catalytic processes mediated by silver complexes and silver nanoparticles in all catalysis domains, including photocatalysis with important consequences also in processes of the uttermost interest such as $CO_2$ photocatalytic reduction to methane.

Third, far from being laboratory curiosities, silver-based MORALs and SACs are catalytic materials of practical and industrial relevance due to the simple and highly reproducible synthetic routes and to the technical and economic advantages offered by their high selective activity and enhanced chemical stability, which directly translates into lower operational costs.

Fourth, being generally comprised of stable metallic species easily deposited onto high surface area stable oxides such as silica, titania, and alumina, both MORALs and SACs can be easily applied to continuous processes carried out in flow reactors of direct relevance to the fine chemical industry for the production of complex molecules, including pharmaceutical ingredients.

Fifth, awareness of the knowledge about these materials amid today's young researchers and scholars remains elusive, new courses in catalysis science and technology should devote direct and significant attention to these new catalytic materials. Some of us have suggested elsewhere how this can be done in an effective way adopting a system thinking approach [33]. Such education will encompass green chemistry and the need to incorporate a closed-loop approach to new catalyst development for which silver recovery and recycling is incorporated since the early phase of the new catalytic material development. Like most scholars, chemistry and chemical engineering undergraduate and graduate students publishing in the field of silver catalysis, have little knowledge of the issues concerning silver production, use, recovery, and recycling. For example, ask one of your aforementioned colleagues the following two questions: "How much silver is used in industrial catalysts?" or "How much silver is recovered from ethylene oxide plants?"

Adopting the aforementioned systems viewpoint, these doubts can be easily dispelled. Students, for example, are taught that EO catalysts typically contain an overall amount of over 3100 tons of silver (deposited on alumina) spread across some 400 plants, half of which are located in three countries only (China, Saudi Arabia, and USA) [34]. In total, around 1500 tons of silver are derived each year globally from spent EO catalysts, with a frequency of recycling dictated by the typical shut down and catalyst change occurring at EO plants every 2.5 years [34].

Sixth, research on new generation silver-based catalysts based on silver MORALs and SACs should target massive petrochemical productions leading to large amounts of undesirable waste such as, for instance, that of propylene oxide. Previous research dating back to 2010 indicated that Ag$_3$ trimers selectively catalyze propylene oxide formation, avoiding combustion to $CO_2$ [35]. Will that be the case also for new generation silver SACs? Is it possible that new silver MORALs such as that

enhancing the selectivity of methanol conversion to formaldehyde will do the same also for propylene oxide production?

Seventh, awareness that sustainability megatrends are currently reshaping the chemical industry [36], both research and education on new generation silver-based catalysts should also focus on the selective conversion of biomass-derived substrates into high value-added building blocks, as happens with Ag/ZrO$_2$ exclusively producing 5-hydroxymethyl-2-furancarboxylic acid (HFCA) via the aerobic oxidation of 5-(hydroxymethyl)furfural (HMF) in a wide range of reaction parameters in almost quantitative yields with a maximum productivity of 400 mol HFCA h$^{-1}$ mol Ag$^{-1}$ [37].

Getting back to the aforementioned systems thinking approach to education in the university considered as a system whose purpose is defined from the perspective of students—users and customers of university teaching ("*provide me with all the facilities and help I need to achieve a positive outcome from my time at your university*") [38], new generation silver-based catalysis is a useful tool to be used in new educational and chemistry research programs around what matters to today's students and tomorrow's chemistry professionals. This study offers examples, ideas, and tools with which to start said new programs focusing on silver catalysis, whose applications and practical impact will soon approach those of gold.

**Author Contributions:** Conceptualization, R.C., C.D.P., and M.P.; writing—review and editing, R.C., C.D.P., M.P., and F.M. All authors have read and agreed to the published version of the manuscript.

**Funding:** This research received no external funding.

**Conflicts of Interest:** The authors declare no conflict of interest.

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
