# Peer review of "Catalysis with Silver: From Complexes and Nanoparticles to MORALs and Single-Atom Catalysts"

_catalysts, doi:10.3390/catal10111343_

Round 1

Reviewer 1 Report

This manuscript organizes catalysis performance of silver metal‐organic alloys and single‐atom catalysts. The authors do a pretty good job to collect the related examples and theory. But there are still some suggestions needed to be revised before publication.

  1. The authors say “The presence of entrapped phthalocyanines does not influence the position and shape of typical lines in the XRD diffractogram with similar patterns obtained both for the doped and undoped Ag nanoparticles.” in line 143-145. Do you have further evidence to show this result?
  2. In Table 1, the second catalyst “Rh/Al2O3” has a missing light source.
  3. Some figures need to be modified. The resolution of Scheme 1, Fig. 1 and Fig. 2 should be higher. These figures are quite blurred and hard to read.

Author Response

Response to Reviewer 1 Comments

Point 1: The authors say “The presence of entrapped phthalocyanines does not influence the position and shape of typical lines in the XRD diffractogram with similar patterns obtained both for the doped and undoped Ag nanoparticles” in line 143-145. Do you have further evidence to show this result?

Response 1: We thank the Reviewer for giving us the opportunity to deepen this result. In the original paper (ref. 25), a XRD diffractogram of undoped Ag is reported where all the peaks attributable to the metal are highlighted (2Φ = 38.28, 44.48, 64.68, 77.68, 81.88, 97.98). Although no diffractogram of doped Ag is reported in that paper, the authors state that similar patterns were obtained for both the doped and undoped metal. They also determined primary particle size by the Scherrer’s equation applied to all the six peaks leading to an average value of 11 nm, both for doped and undoped silver. Furthermore, SEM images showed that the morphology of the powders appeared to be the same for both doped MORALs and undoped silver with aggregates of hundreds of nanometers up to few micrometers. We have added these information in lines 154-158.

Point 2: In Table 1, the second catalyst “Rh/Al2O3” has a missing light source.

Response 2: We are grateful to the Referee for asking us to correct Table 1 properly. We have added the missing light source (LED) related to Rh/Al2O3 catalyst

Point 3: Some figures need to be modified. The resolution of Scheme 1, Fig. 1 and Fig. 2 should be higher. These figures are quite blurred and hard to read.

Response 3: Following the Reviewer’s suggestion, we have modified the figures and schemes in order to improve their resolution and/or making them different from the original source.

Reviewer 2 Report

This manuscirpt reviewed the silver-based catalysts from metal complex/nanoparticles to metal organic alloys and single atom catalysts. The development and advantages of those catalysts are outlined. A few recommendations are listed below for revision. 

  1. More references are recommended for a review. In the introduction, more references are recommended when a conclusion or summary was made. For example, the first paragraph in the introduction described a broad picture of silver catalysts for synthetic organic chemistry. there should be more than one references to be cited. Also, there are many places without reference to support. For example, int the first sentence of second paragraph, “Silver(i) is a ….. multiple bonds” was stated without references to support. The introduction with current structure is also hard to follow as well. 
  2. In section 2 and 3, the author focused on the metal organic alloys and silver single atom catalysts. There were long paragraphs talking about the synthetic details, which is less important here. The key focus should be the advantage of those materials compared with other materials and why those materials can achieve better catalytic performance. Furthermore, more reference should be included.
  3. There is only one table is about catalysis. the rest of figures are all about materials. It is recommended to reorganize the figures, which can show the improvement and advantages in catalysis using those silver-based catalysts

Author Response

Response to Reviewer 2 Comments

Point 1: More references are recommended for a review. In the introduction, more references are recommended when a conclusion or summary was made. For example, the first paragraph in the introduction described a broad picture of silver catalysts for synthetic organic chemistry. there should be more than one references to be cited. Also, there are many places without reference to support. For example, in the first sentence of second paragraph, “Silver(i) is a ….. multiple bonds” was stated without references to support. The introduction with current structure is also hard to follow as well.

Response 1: We thank the Reviewer for his/her proper suggestion. Further references have been added in the Introduction and throughout the text (reported in red). Furthermore, we have stressed the importance of minimizing metal size from bulk to single-atom passing through nanometric scale in lines 60-69.

Point 2: In section 2 and 3, the author focused on the metal organic alloys and silver single atom catalysts. There were long paragraphs talking about the synthetic details, which is less important here. The key focus should be the advantage of those materials compared with other materials and why those materials can achieve better catalytic performance. Furthermore, more reference should be included.

Response 2: Sections 2 and 3 were intended to describe the synthetic procedure and catalytic performance of some MORALs and SACs taken as representative. A focus on the key advantages offered by these catalysts and consequent research and educational guidelines was reserved to section 4. In line 54-55 and at the end of the Introduction, it was explained the scope of this short review: “In this study, we focus on selected recent advances concerning processes mediated by new metal organic alloys (MORALs) and silver single-atom catalysts (SACs)” and “rather than providing a comprehensive review of all achievements in catalysis with these new materials, we offer a critical insight aimed at suggesting avenues to guide the development of new single-atom and silver-organics alloyed catalysts. would focus on selected recent advances”.

Point 3: There is only one table about catalysis. The rest of figures are all about materials. It is recommended to reorganize the figures, which can show the improvement and advantages in catalysis using those silver-based catalysts

Response 3: Following the Referee’s suggestion, we have added a Scheme (Scheme 1) and omitted two Figures (former Figs. 2 and 4). Furthermore, we have modified some figures and schemes in order to improve their resolution and/or making them different from the original source. 

Round 2

Reviewer 2 Report

In scheme 1, the author presented several microscope images. it is recommended to add the reference in the caption if they are from literatures. 

I have no further comments. 

Author Response

Point 1: In scheme 1, the author presented several microscope images. it is recommended to add the reference in the caption if they are from literatures.

I have no further comments.

Response 1: The Referee’s recommendation is right, we thank him/her. Actually we did not add any reference in the caption because those images are taken and modified not exactly from the literature but from an informative article in the web at https://www.lastampa.it/tuttoscienze/2020/04/08/news/maescherine-e-spray-a-base-di-nanoparticelle-1.38693951. Following his/her suggestion we have added this link in the caption (if proper).